# Rapid Test Method for Evaluating Inhibiting Effectiveness of Supplementary Cementitious Materials on Alkali–Silica Reaction Expansion of Concrete

**DOI:** 10.3390/ma15093202

**Published:** 2022-04-28

**Authors:** Lei Yi, Zhongyang Mao, Min Deng, Xiang Liu, Zhiyuan Fan, Xiaojun Huang, Tao Zhang, Mingshu Tang

**Affiliations:** 1College of Materials Science and Engineering, Nanjing Tech University, Nanjing 211800, China; 201961203130@njtech.edu.cn (L.Y.); mzy@njtech.edu.cn (Z.M.); 201961203dengmin@njtech.edu.cn (M.D.); 201961203150@njtech.edu.cn (X.L.); 201961203172@njtech.edu.cn (Z.F.); 5967@njtech.edu.cn (X.H.); 201961203162@njtech.edu.cn (T.Z.); 2State Key Laboratory of Materials-Oriented Chemical Engineering, Nanjing 211800, China

**Keywords:** alkali–silica reaction, concrete prism test, temperature, accelerated inhibition test, aggregates

## Abstract

At present, there are many problems in various tests when judging the alkali activity of aggregates. The most practical engineering concrete prism test (CPT) takes one year, and the concrete suppression method needs two years. The aim of this paper is to discuss inhibiting effectiveness of supplementary cementitious materials (SCMs) on alkali–silica reaction (ASR) expansion of concrete and evaluate this rapid test method. Three kinds of aggregates were selected by chemical analysis, XRD and petrographic analysis. The high alkali–silicic acid activity of three aggregates was determined by accelerated mortar bars, concrete microbars and CPT. The expansion of concrete specimens made of three kinds of aggregates was measured and analyzed by using the method of length measurement. By changing the curing temperature to 40 °C, 60 °C and 80 °C, the test period of CPT is accelerated. It proved that the expansion of CPT is larger at 60 °C and smaller at 40 °C. The inhibition test was also accelerated by adding different proportion of SCMs (fly ash or blast furnace slag) and adjusting the curing temperature to 60 °C and 80 °C. On this basis, the inhibition test was accelerated by changing NaOH solution instead of moist air curing. The test period of the accelerated inhibition test can be effectively shortened from two years to 4 months, The expansion trend of each parameter and specimen is evaluated, the evaluation cycle can be adjusted to 5–6 months. The microscopic reaction characteristics of concrete specimens were investigated by means of SEM. According to each parameter and criterion, the judging standard of concrete rapid test and rapid restraint test is given in this paper.

## 1. Introduction

AAR is generally divided into alkali–silica reaction (ASR) [1,2] and alkali–carbonate reaction (ACR) [3,4]. It refers to the chemical reaction between some active minerals in aggregate and alkali in concrete micro-pores [5], which results in internal self-expansion stress and cracking [6]. In the past half century, many concrete projects have suffered a bitter lesson and caused huge economic losses [7]. Given that the reaction takes place inside the concrete [8], it is non-stop and difficult to repair; thus, it can only be prevented [9].

ASR is a process that the alkali in concrete reacts with the active silica in aggregate to form alkali–silica gel. The gel expands after absorbing water, which leads to the cracking and destruction of concrete [10]. It is generally believed that three factors must be met for ASR to occur: alkali, active aggregate and water [11]. At present, the main measures to prevent ASR are: (1) use of non-active aggregate. (2) control of alkali content in concrete [12,13]. (3) control of humidity in surrounding environment. (4) use of mixed materials or chemical admixtures [14].

At present, adding SCMs into concrete had become the most common measure to restrain ASR expansion [15]. It can not only restrain ASR expansion to some extent, but also improve other properties of concrete [16], such as fly ash, silica fume, slag powder, metakaolin, etc. [17,18].

The presence of reactive aggregate is a necessary condition for ASR to occur. The scientific communities avoid using reactive aggregate, but due to the shortage of resources and regional restrictions, some reactive aggregates have to be used. At this time, to strengthen the detection of aggregate alkali activity has become the key to prevent ASR expansion. The research of aggregate alkali activity evaluation method is an important aspect in the ASR research field [19]. Many studies have been carried out on the evaluation methods of aggregate alkali activity. There are chemical methods to identify the composition of aggregate, mortar bar method based on length measurement, concrete prism test, autoclave method, etc. [20]. These methods are constantly verified and improved in practice, countries eventually developed standards and codes to prevent AAR [21], such as Canada’s CSA standard, the United States ASTM standard [22,23,24,25,26] and the international RILEM standard [27].

In this paper, three kinds of rocks are used, the metamorphic rocks are Jiuli slate (recorded as JL), the sedimentary rocks are Lianghekou sandstone (recorded as LHK) and the magmatic rocks are Ningbo Tuff (recorded as NHY). To explore the composition and internal structure of the three types of rocks [28]. Chemical analysis, XRD analysis and petrographic analysis were used to prove that they are consistent with silicate aggregates [29]. Accelerated mortar bars [23], concrete microbars [27] and CPT [22] are used to prove that they have high alkali silicic acid activity. Three types of rock are made into concrete specimens according to the standards [30]. Then, they were cured in moist air at 40 °C, 60 °C and 80 °C, respectively [31]. The expansion of concrete specimens is measured periodically to discuss the influence of curing temperature on the expansion of concrete specimen. By comparing with the data of CPT, it was put forward to accelerate the evaluation of test period of CPT [32,33]. On this basis, the paper added different SCMs (FA, BFS), changed the curing temperature and curing medium to accelerate the inhibition period. Through the expansion data analysis, the accelerating judging basis and judging method are put forward [34]. During the inhibition test, the basicity of the curing solution was measured. The paper made a reasonable comparison between the alkali consumed and the degree of reaction. At last, the new interface of reactive aggregate was analyzed by SEM in order to explore the formation of alkali–silica gel product and crack. In this paper, shortening the test cycle of concrete prism test and restraining test will serve the practical engineering more efficiently.

## 2. Materials and Methods

### 2.1. Cement and Aggregates

The cement was selected as P·II 52.5 Portland cement produced by Jiangnan Onoda Cement Co., Ltd. (Nanjing, China). Its equivalent alkali content is 0.56%. Table 1 shows the chemical composition of cement.

All the aggregates used in the test are applied in practical engineering field. The appearance of the aggregates is shown in Figure 1. They are Guizhou JL slate, Sichuan LHK sandstone and Zhejiang NHY tuff.

The chemical composition of aggregates was determined according to GB/T 176-2017 (CIS, 2017). The results of chemical composition analysis are shown in Table 2.

XRD was performed on the D8 Advance diffractometer (Bruker AXS GmbH, Karlsruhe, Germany). The mineral composition of sandstone was qualitatively analyzed by X-ray diffractometer. The experimental conditions were as follows: Cu Target, 40 kV, 40 Ma, scanning speed 5°/min, step length 0.005°. The mineral compositions are shown in Figure 2. JL slate mainly consists of quartz with a small amount of calcite, muscovite and chlorite, whereas LHK sandstone mainly consists of quartz, chlorite, illite and albite; the main mineral phases of NHY slate are quartz, chlorite and albite. The Rietveld analysis data are shown in Table 3, which show that all aggregates contain a significant amount of Quartz and a certain amount of other minerals.

Microstructure characteristics of samples were performed by means of optical microscopy (Light Microscopy Zeiss LSM 800, Jena, Germany). The lithofacies of the thin plane prepared from three aggregate samples have been observed by Polarized light microscopy method [25]. As shown in Figure 3, the JL slate is mainly composed of mica and microcrystalline feldspar, containing a small amount of calcite, iron ore, chlorite and about 4% of microcrystalline quartz. Microcrystalline quartz is a potential alkali active component.

The microstructure of the LHK sandstone is shown in Figure 4. The main mineral phases are quartz crystal, feldspar crystal and mica, and also contain a small amount of calcite and about 10% of microcrystalline quartz.

The microstructure of NHY tuff is shown in Figure 5. The main mineral phases are quartz crystal and feldspar crystal. The phenocrysts have a speckled structure, the matrix is microcrystalline feldspar and a little mica, chlorite hornflash and about 6% microcrystalline quartz.

The fly ash used in the concrete sample is Fuxi grade I FA. The density is 2.4 g/cm^3^ and the equivalent alkali content is 1.68%. The BFS is produced by Shanghai Baosteel New Building Materials Technology Co., Ltd. (Shanghai, China), and the chemical composition is shown in Table 4. The water reducing agent is produced by Jiangsu Subote Co., Ltd. (Nanjing, China). The water reducing rate is 22% and the alkali content is 1.49%. The NaOH used in the experiment is a chemical reagent.

### 2.2. Accelerated Expansion Test of Concrete

It is necessary to measure the expansion of concrete specimen, and the formula is shown as (1):P_t_ = (L_t_ − L_0_)/(L_0_ − 2b) × 100%(1)
where P_t_ is the expansion after t days of curing, in %; L_t_ is the test piece length after t days of curing, in mm; L_0_ is the initial test piece length, in mm; and b is the length of the nail embedded in the concrete, in mm.

The alkali–silicic acid reactivity of three test aggregates was determined by the ASTM C1260, ASTM C1293 and RILEM AAR-2. The expansions of alkali–silicic acid reaction were determined, as shown in Figure 4.

As shown in Figure 6a, the expansions of three kinds of aggregate mortar samples at 14 days age reached 0.378%, 0.391% and 0.36%, respectively, and 0.20% is the expansion limit of mortar bar rapid method samples. According to Figure 6b, the expansions of three kinds of aggregate mortar samples at 28 days age reached 0.164%, 0.271% and 0.226%, respectively, far exceeding 0.10%. As shown in Figure 6c, at the age of 52 weeks, the expansions of three kinds of aggregate mortar samples reached 0.074%, 0.054% and 0.069%, respectively, far exceeding 0.04%. In summary, all the aggregates (JL, LHK, NHY) were determined to have alkali–silicic acid reactivity.

On the basis of concrete prism method, the concrete specimens of 75 mm × 75 mm × 285 mm are formed by five-gradation of the aggregates. The alkali content of concrete is 5.25 kg/m^3^ by adding NaOH and mixing water. The concrete specimens were cured at 40 °C, 60 °C and 80 °C in moist air 24 h after forming [35]. The influence of temperature on the expansion was explored [36]. The change in curing temperature does not adversely affect the aggregates or other constituents’ properties obviously [37].

### 2.3. Accelerated Concrete Inhibition Test

When the concrete sample is formed, the supplementary cementitious materials are added to replace the cement, which can restrain the ASR expansion of the concrete samples [38]. They also formed by five-gradation of the aggregates, and they were cured at 60 °C and 80 °C in moist air 24 h after forming. For the accelerated test of concrete inhibition, this method adopted 1.5 mol/L NaOH solution curing at 60 °C and 80 °C. By changing the curing temperature and curing medium of concrete specimen, the aim of accelerated inhibition test is achieved [39]. The relationship between curing temperature and curing medium for accelerated inhibition test is put forward [30,40].

## 3. Results

### 3.1. Effect of Temperature on Expansion

Whether different curing temperature can change the expansion rate of ASR or not, and whether it has an effect rule on ASR expansion [41]. In this chapter, the test based on CPT but changed the curing temperature of specimens to accelerate the test time of CPT and shorten the time of evaluating alkali aggregate activity. Under the condition of changing curing temperature, it is necessary to ensure the control of molding mix ratio, alkali equivalent, particle gradation and water–binder ratio. Any small change in the parameters will lead to a large error in the final test results.

In order to solve the above problems, the concrete samples were molded with coarse and fine aggregates of three kinds of rocks, respectively, and the parameters were adjusted to reach the same. The influence of curing temperature on the ASR expansion of concrete was investigated by measuring the expansion value of the specimens.

Figure 7a–c show the expansion of specimens with three kinds of aggregates at different temperatures. As shown in Figure 7a, the expansion of the three kinds of aggregate mortar specimens with curing temperature of 40 °C was more than 0.04% at 16 weeks age, the mortar specimens with curing temperatures of 60 °C and 80 °C expanded more rapidly in the 0–16 weeks age. At three different temperatures, the expansion trend of concrete specimens tends to be the same. In the early stage, the expansion at 80 °C is the fastest, exceeding the limit value of 0.04% at the age of 4~5 weeks. However, the expansion tends to slow down, and there is little change after half a year. At 40 °C and 60 °C, the expansion is slightly slower than at 80 °C in the early stage, but the expansion at 60 °C still occurs at the later stage.

With the increase in temperature, the expansion of prismatic concrete specimens with three kinds of aggregates will be larger, but it does not conform to the rule that the higher curing temperature is, the bigger the ASR expansion is. The concrete prism specimens of three kinds of aggregates all showed the maximum expansion at 60 °C and the minimum expansion at 40 °C. The deformation of concrete sample is due to ASR expansion and hydration shrinkage of cement paste. In the early period of curing, the self-shrinkage of concrete samples is more serious than ASR expansion because of hydration of cement paste, it is practical to measure the negative value of expansion when the concrete specimen shrinks. As time goes on, the ASR reaction intensifies, and the mid-term concrete specimens also show considerable expansion. At the later stage of curing, the expansion of three kinds of aggregate concrete prisms tends to be smooth and reach a relatively stable value under three kinds of temperature.

### 3.2. SCMs Inhibits ASR

Studies have shown that adding 25% FA, 60% BFS or composite doping can effectively inhibit the occurrence of ASR expansion. Therefore, the ratio used in this test can have a better inhibition effect on ASR. This paper only discusses the change in curing temperature and curing medium to accelerate the process of inhibiting ASR effectively. Table 5 shows the content required for various materials to be used in concrete specimens. The experiment was divided into six experimental groups and one control group. The mixture of 20% and 40% FA, 40% and 60% BFS and 40% and 60% mineral powder (half of FA and BFS) were used. The cement content is reduced according to the addition of SCMs, and the water–cement ratio keeps consistent. In order to prevent the alkali penetration of additives, the water-reducing agent is added as 4.8 g, which not only made the sample easier to shape, but also kept the alkali content of each group the same [22].

Figure 8, Figure 9 and Figure 10 showed the expansion of concrete prism specimens mixed with SCMs at curing temperatures of 60 °C and 80 °C. As can be seen from the figures, the expansion value of the specimens with six different supplementary cementitious materials did not reach the critical value of 0.040% in one year, whereas the standard was that the expansion value did not exceed 0.040% in two years. Taking into account the longer period, and the expansion value of the sample tends to be stable in the range of 26~52 weeks; therefore, one year is taken as the reference standard here. The addition of 20% FA and the addition of 40% BFS have relatively high expansion and weak inhibition compared with composite doping and high proportion doping. The composite doping has a relatively good performance in the inhibition. In the early stage of curing, some of the specimens did not expand. In consideration of the measurement error and the self-shrinkage of the early cement, the expansion of the concrete specimens tended to smooth during the whole period, there has been no dramatic increase.

Comprehensive analysis of the above test results, the addition of 20%, 40% fly ash, 40%, 60% blast furnace slag, and 40%, 60% compound doping SCMs all had good inhibition effects on ASR expansion. The expansion of concrete specimen is mostly in the range of 0.002%~0.015% in one year. Because of the time of the experiment, the following 1–2-year provisional period was not measured, but the curve of the expansion showed that the expansion did not increase significantly in the following 1-year provisional period, the expansion curve has become smooth; therefore, the expansion of the concrete sample will not reach 0.04% in 2 years. The change in curing temperature affects the reaction of alkali–silica reaction to a certain extent, but it does not affect the inhibition effect of SCMS on ASR.

### 3.3. Accelerated Inhibition Test

It takes two years for the concrete inhibition test, and the interim period is much longer than people expected. Therefore, we hope that the interim period can be shortened by changing the temperature or curing condition to accelerate the inhibition test. The curing temperature is 60 °C or 80 °C in 1.5 mol/L NaOH alkali solution. Figure 11, Figure 12 and Figure 13 show the expansion of specimens with three kinds of aggregates and SCMs cured in 60 °C or 80 °C, 1.5 mol/L NaOH alkali solution.

The concrete specimens prepared with three kinds of aggregates had larger expansion than that cured by moist air, which were mixed with different proportion of SCMs. By increasing the reaction temperature and pressure, the reaction speed is accelerated, and the evaluation time is shortened. It is not difficult to see that there is a certain correlation between the rapid inhibition test data and the moist air curing data. For the JL slate aggregate, when curing at 60 °C, the expansion of alkaline solution curing at 8 to 16 weeks is similar to that of moist air curing at half a year to one year, the expansion of alkali solution curing at the beginning of the sixth week was similar to that of moist air curing at the 16th week, and the trend of expansion was similar too. The same is true for the LHK sandstone and NHY tuff aggregates. It can be concluded that the test cycle of accelerated inhibition test for alkali solution can be effectively shortened from the standard two-year inhibition period to four months. The judging period can be extended to five to six months caused that the samples may still grow at the later stage of the accelerated inhibition test, but the increasing trend is slow and steady.

## 4. Discussion

### 4.1. Alkalinity Reduction Test

During the inhibition test, the OH^-^ in the NaOH solution (initial concentration 1.5 mol/L) was continuously consumed with the ASR, and the alkalinity of the reaction system decreased at each temperature as shown in Figure 14. With the ASR, the basicity of the reaction system tends to increase. The alkalinity at the early stage (3 days) of the reaction decreased significantly but then the alkalinity decreased (the cause has yet to be found). It started to grow again but it did not grow much in the later period. We should be noted that the concentration of alkali in the system remained at a relatively high level of 0.8 mol/L even when the reaction was slowed down.

According to the changing trend in NaOH solution basicity reduction value at 40 °C, 60 °C and 80 °C, it can be concluded that there was a positive correlation between basicity hydroxide radical reduction value and ASR gel formation degree at each temperature at the beginning of reaction. As the reaction continues, the ASR gel content inside the particles increases continuously. Under the expansion pressure of ASR gel, the gel will accelerate the diffusion and dissolution to the alkali solution outside, and the higher the temperature, the greater the diffusion coefficient. In the late reaction period, the system remained relatively balanced, the ASR gel content increased slowly, and the reaction rate decreased significantly.

### 4.2. Microscopic Analysis

In this section, the microscopic reaction characteristics of concrete specimens are studied by means of scanning electron microscopy (FE-SEM Zeiss Ultra Plus equipped with EDS Oxford X-Max SDD 50 mm^2^ 106 detector and INCA 4.14 5 X-ray microanalysis software). The selected specimens were LHK specimens cured at 60 °C for 1 year by moist air, LHK specimens mixed with 60% SCMs and cured at 60 °C for 40 weeks by wet air, LHK specimens mixed with 60% SCMs and cured at 60 °C for 12 weeks by 1.5 mol/L NaOH (recorded as sample 1, 2, 3). The aggregates were taken from the three samples and their reaction surfaces were tested by SEM. The micrograph of the concrete specimen is shown in Figure 15. From Figure 15a, it can be seen that a large number of microcrystalline quartzes in the aggregate reacted and the specimen expanded greatly. Cracks have developed in the interior of the aggregate, the white substance was found in the crack. The results of energy spectrum analysis showed that the product was alkali–silica gel. No cracks were observed by naked eye in sample 2, 3. The inhibition effect of SCMs is obvious. The specimen expanded slightly by 0.005%, and there was almost no gel material in the inner pores of the specimen. From Figure 15c,d, a small amount of amorphous ASR gel material was produced at the edge of aggregate, but not a large amount of honeycomb or spherical gel material. The doping of SCMs can change the morphology of the reaction products.

The doping of SCMs not only dilutes the alkali in concrete, but also reacts with the alkali in concrete. It causes the aggregate surface to contain only a small amount of alkali ions and changes the microstructure of the concrete. The above analysis better supports the supposition that the correctness of the method can be accelerated by the change in curing temperature and curing medium.

### 4.3. Determination of Parameters in Accelerated Inhibition Test

ASR expansion of concrete is closely related to the following factors: alkali content (A), alkali activity level (L), ambient temperature (T), relative humidity (H), mix ratio (M), restraint stress (σ) and time (t), etc. Its expansion can be expressed as a general formula as shown in (2):ε = f (A, L, T, H, M, σ, t)(2)

For a particular concrete, the alkali content (A), the alkali activity level (L) and the mix ratio (M) can be regarded as a constant value. If the relative humidity (H) of concrete can be maintained above 85%, the effect of humidity on ASR expansion of concrete can be ignored; For a particular concrete, the confining stress (σ) of concrete generally does not change significantly. As a result, ASR expansion of concrete is more subject to ambient temperature (T) and time (t).

The selection of specimen parameters should be as close as possible to the actual concrete. The selection of specimen size should take into account both the accuracy of measurement and the representativeness of sampling. If the specimen size is too small, it will be too different from the actual concrete and does not have the engineering application ability. If the specimen size is too large, the measurement will be difficult, and the measurement error will be large. The water–cement ratio and cement–sand ratio are as close as possible to the actual concrete project. At the same time, due consideration should be given to the laboratory operability, and the aggregate size and gradation are as close as possible to the actual concrete project. It should match with the size of the specimen. Therefore, the size of the specimen is 75 mm × 75 mm × 285 mm, water–cement ratio is 0.47, sand–cement ratio is 0.25.

By comparing the experimental results under the condition of high temperature and high alkali with the experimental results under the condition of no alkali, under the condition of curing at 60 °C or 80 °C in 1.5 mol/L NaOH solution, there is a good correlation between the expansion of 0.04% in 4 months (extended to 6 months) as a criterion under the condition of rapid growth, and the expansion of 0.04% in one year under the condition of wet moist curing. Whether this criterion can be used as the final SCMs to evaluate the ASR suppression effect of the mixture needs to be verified and improved by a large number of high, medium and low activity aggregates.

## 5. Conclusions

Based on the test results of concrete specimens prepared with three different alkali–silica reactive aggregates mixed with different SCMs and different curing conditions, we can get the following conclusions.

(1).The accelerated concrete inhibition test shows that the test can be effectively shortened from the standard 2-year inhibition period to 4 months. Considering that the sample may still grow in the later period of accelerated concrete inhibition test, but the growth trend is slow and stable, the judgment period can be extended to 5 to 6 months and the expansion limit can be set at 0.04%. For the change in aggregate type, SCMs type and low content of SCMs, whether the criterion of accelerated inhibition test can be used as a reference standard is worth further study.(2).The influence of temperature on the inhibition effect of mixed SCMs is not obvious, and the change in temperature will not accelerate or slow down the inhibition degree to a great extent. However, the effect of temperature on concrete prism test is obvious. The concrete specimens show larger expansion at curing temperature of 60 °C and smaller expansion at 40 °C.(3).There was a positive correlation between basicity hydroxide radical reduction value and ASR gel formation degree. The change in temperature directly affects the formation of ASR gel and the expansion of concrete. Whether the change in curing temperature directly affects the internal structure and product characteristics of concrete still needs a lot of research.

## Figures and Tables

**Figure 1 materials-15-03202-f001:**
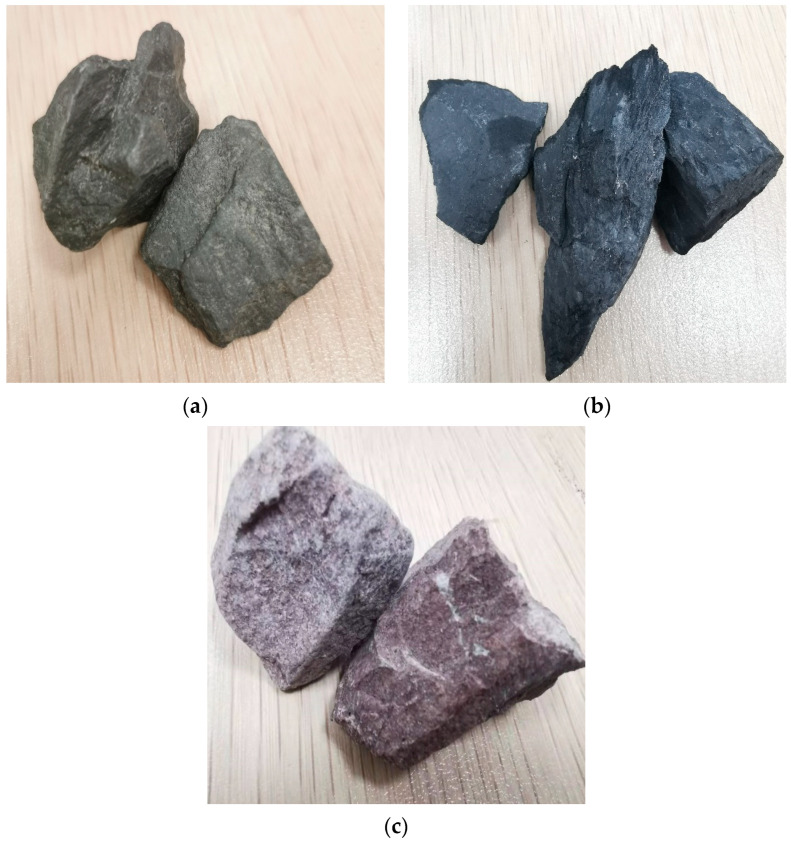
Appearance of aggregates. (**a**) The visual of JL slate; (**b**) The visual of LHK sandstone; (**c**) The visual of NHY tuff.

**Figure 2 materials-15-03202-f002:**
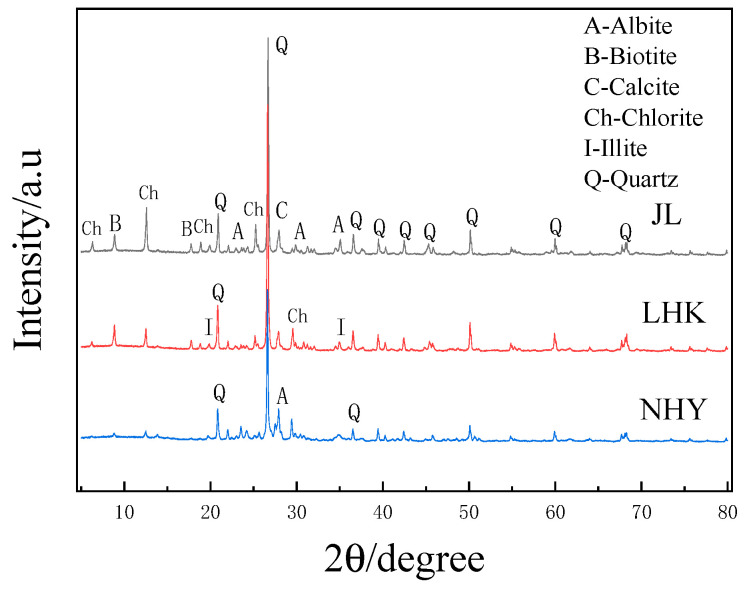
X-ray diffraction pattern of aggregate.

**Figure 3 materials-15-03202-f003:**
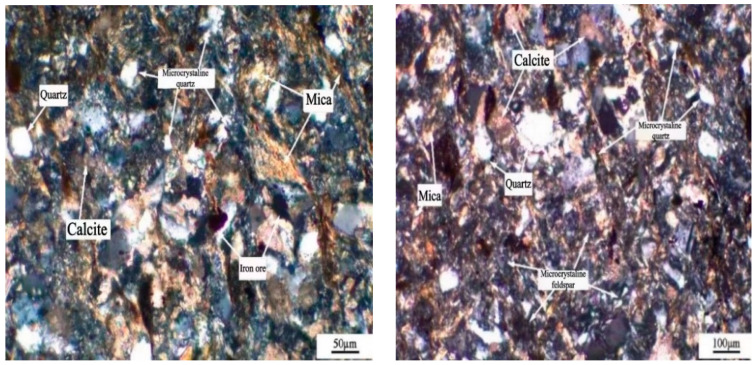
Microstructure of JL slate.

**Figure 4 materials-15-03202-f004:**
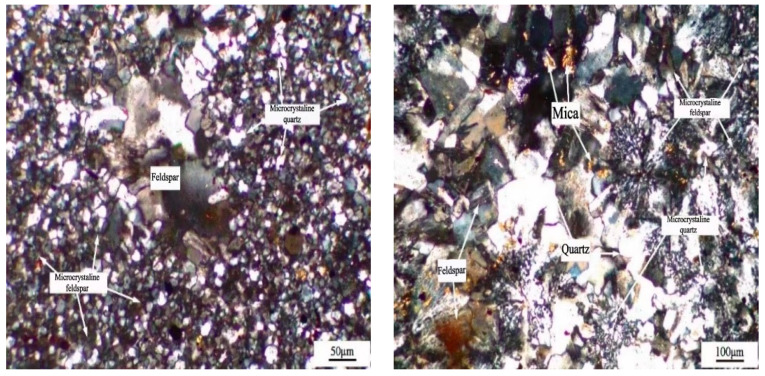
Microstructure of LHK sandstone.

**Figure 5 materials-15-03202-f005:**
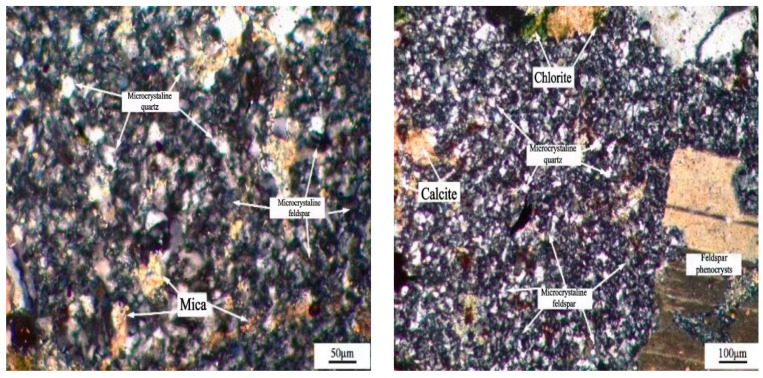
Microstructure of NHY tuff.

**Figure 6 materials-15-03202-f006:**
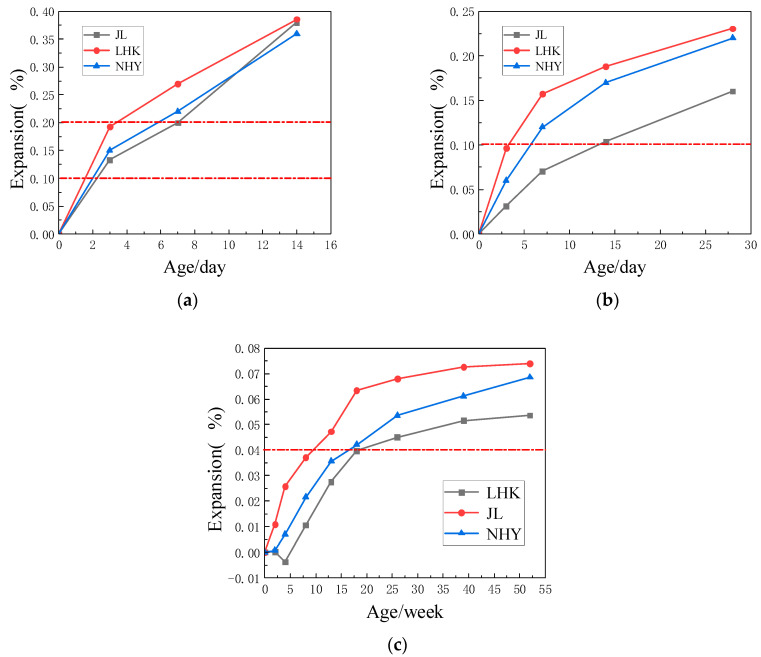
The activity of alkali–silicic acid reaction. (**a**) The expansions of accelerated mortar bars; (**b**) The expansions of concrete microbars; (**c**) The expansions of concrete prisms.

**Figure 7 materials-15-03202-f007:**
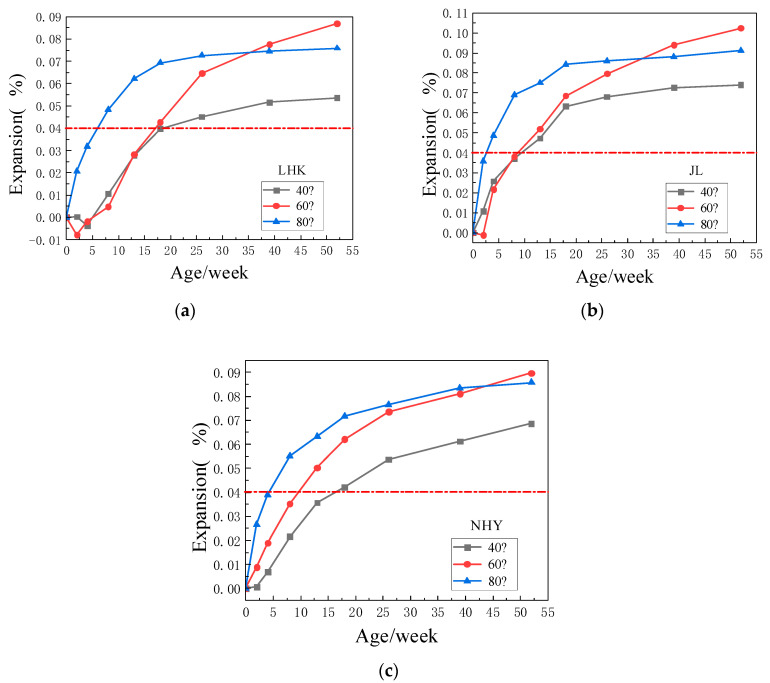
Expansion of specimens with three kinds of aggregates at different temperatures. (**a**) Expansion of specimens with JL slate; (**b**) Expansion of specimens with LHK sandstone; (**c**) Expansion of specimens with NHY tuff.

**Figure 8 materials-15-03202-f008:**
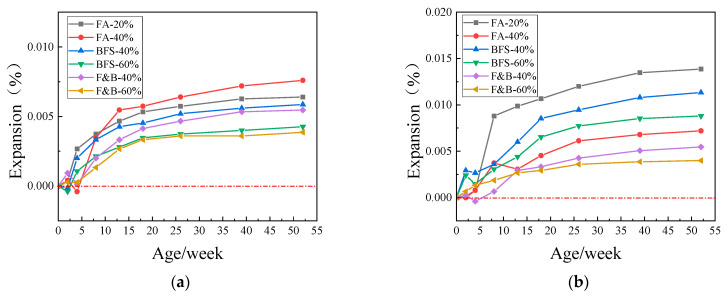
Expansions of concrete prisms with JL slate and SCMs cured in different temperatures. (**a**) 60 °C; (**b**) 80 °C.

**Figure 9 materials-15-03202-f009:**
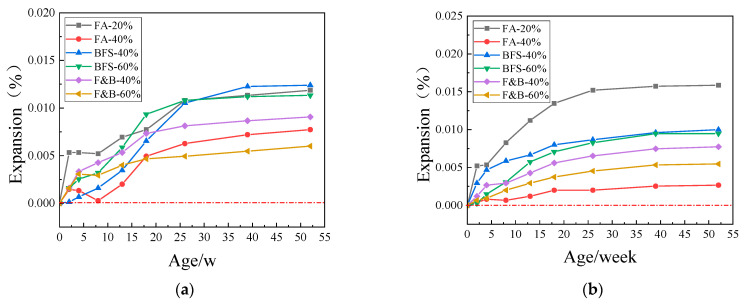
Expansions of concrete prisms with LHK sandstone and SCMs cured in different temperature. (**a**) 60 °C; (**b**) 80 °C.

**Figure 10 materials-15-03202-f010:**
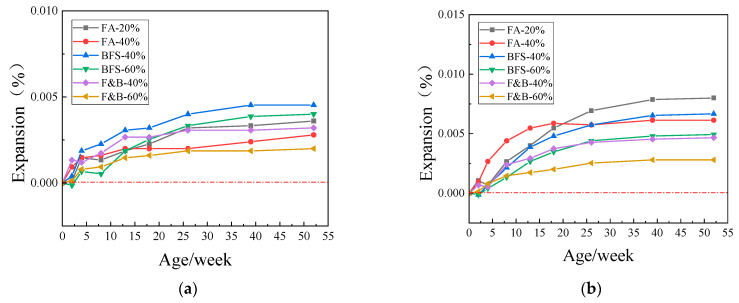
Expansions of concrete prisms with NHY tuff and SCMs cured in different temperature. (**a**) 60 °C; (**b**) 80 °C.

**Figure 11 materials-15-03202-f011:**
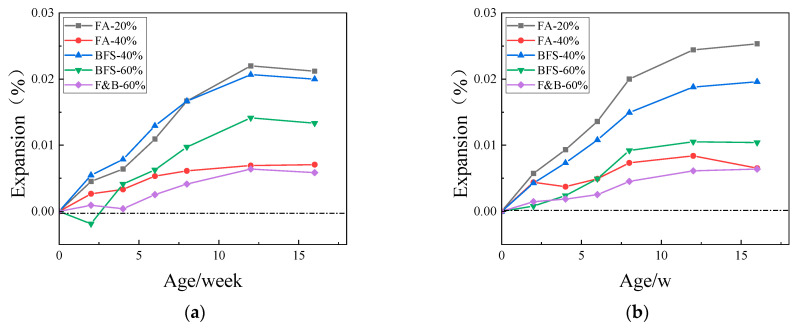
Expansions of concrete prisms with JL slate and SCMs cured in 60 °C or 80 °C, 1.5 mol/L NaOH alkali solution. (**a**) 60 °C; (**b**) 80 °C.

**Figure 12 materials-15-03202-f012:**
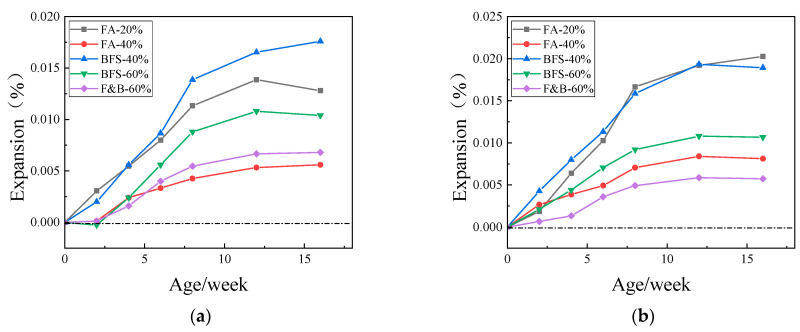
Expansions of concrete prisms with LHK sandstone and SCMs cured in 60 °C or 80 °C, 1.5 mol/L NaOH alkali solution. (**a**) 60 °C; (**b**) 80 °C.

**Figure 13 materials-15-03202-f013:**
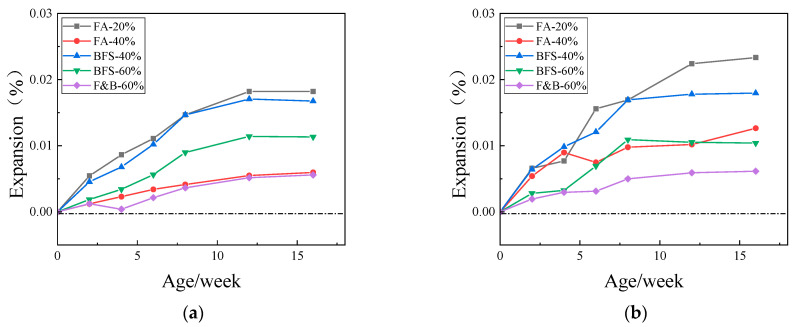
Expansions of concrete prisms with NHY tuff and SCMs cured in cured in 60 °C or 80 °C, 1.5 mol/L NaOH alkali solution. (**a**) 60 °C; (**b**) 80 °C.

**Figure 14 materials-15-03202-f014:**
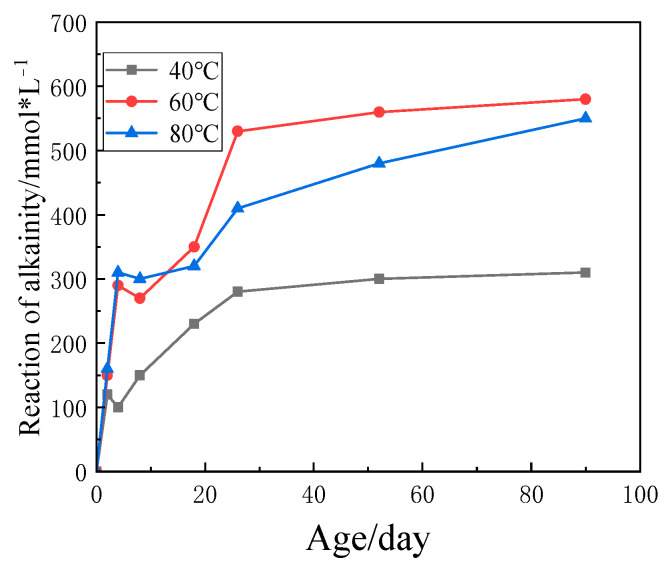
Reduction in alkalinity of NaOH solution with the initial concentration of 1.5 mol/L.

**Figure 15 materials-15-03202-f015:**
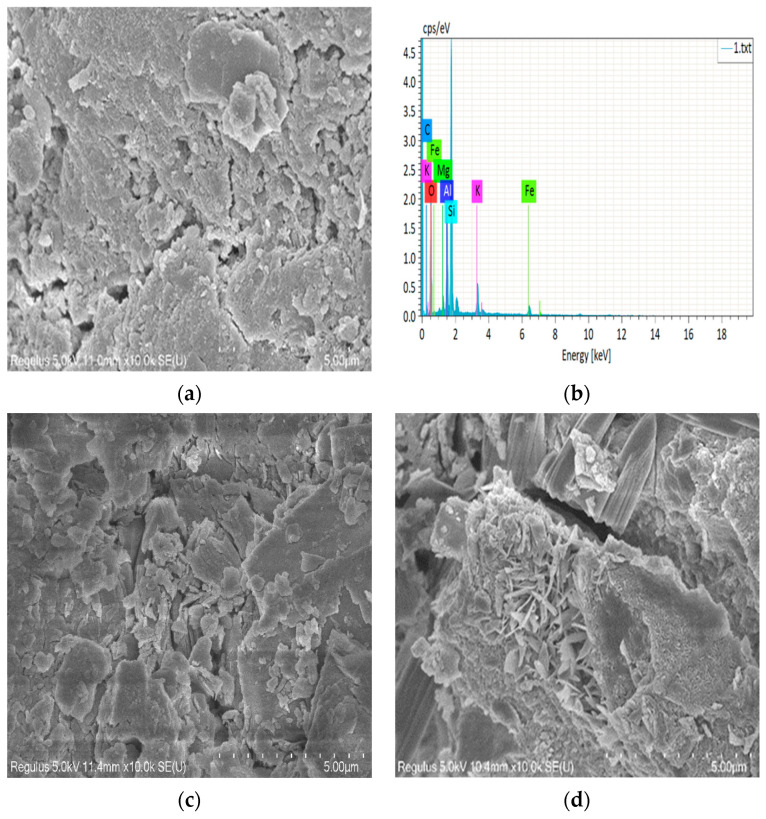
The micrograph of concrete specimens by three testing methods. (**a**) concrete prism test (52 weeks); (**b**) EDS analysis of reaction products at crack; (**c**) Concrete inhibition test (40 weeks); (**d**) Accelerated concrete inhibition test (12 week).

**Table 1 materials-15-03202-t001:** Chemical composition of the cement.

Material	Chemical Composition/wt.%									
	SiO_2_	Al_2_O_3_	Fe_2_O_3_	CaO	MgO	SO_3_	K_2_O	Na_2_O	Loss	Total
cement	19.56	4.28	4.17	63.61	1.27	2.05	0.53	0.21	3.19	98.87

**Table 2 materials-15-03202-t002:** Chemical composition of the sandstones/%.

Aggregates	Chemical Composition/wt.%								
	SiO_2_	Al_2_O_3_	Fe_2_O_3_	CaO	MgO	K_2_O	Na_2_O	Loss	Total
JL	61.00	16.62	8.51	1.75	3.16	3.35	1.50	3.73	99.82
LHK	64.35	15.87	5.31	1.24	1.91	2.82	1.55	3.81	97.00
NHY	72.35	2.26	13.02	1.14	0.25	4.39	2.64	1.13	97.18

**Table 3 materials-15-03202-t003:** Mineralogical composition of the sandstones/%.

Aggregates	Mineralogical Composition/wt.%					
	Quartz	Albite	Illite	Chlorite	Biotite	Calcite
JL	53.5	15.5	17.2	9.1	3.5	1.2
LHK	57.7	13.0	16.9	4.6	3.0	4.8
NHY	56.9	11.6	21.8	0.4	2.3	7.1

**Table 4 materials-15-03202-t004:** Chemical composition of fly ash and blast furnace slag.

Raw Materials	Chemical Composition/%								
	SiO_2_	Al_2_O_3_	Fe_2_O_3_	CaO	MgO	K_2_O	Na_2_O	Loss	Total
FA	49.47	20.44	15.47	5.33	1.25	1.59	0.62	1.71	95.88
BFS	33.52	14.43	0.76	39.48	7.57	0.29	0.42	0	96.47

**Table 5 materials-15-03202-t005:** Materials ratio and SCMs content.

Number	SCMs/%	Materials/g					
		Cement	FA	BFS	NaOH	Water	Water Reducing
1	0	2290	0	0	20.9	1000	4.8
2	20	1832	458	0	18.47	1000	4.8
3	40	1374	916	0	13.85	1000	4.8
4	40	1374	0	916	13.85	1000	4.8
5	60	916	0	1374	9.23	1000	4.8
6	20/20	1374	458	458	13.85	1000	4.8
7	30/30	916	687	687	9.23	1000	4.8

## Data Availability

The data presented in this study are available on request from the corresponding author.

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
