# Peer review of "Rapid Test Method for Evaluating Inhibiting Effectiveness of Supplementary Cementitious Materials on Alkali–Silica Reaction Expansion of Concrete"

_materials, 2022, doi:10.3390/ma15093202_

Round 1
Reviewer 1 Report
Review Assignment for Materials-1667813:
This study "Rapid test method for evaluating inhibiting effectiveness of supplementary cementitious materials on ASR expansion of concrete" as an analytical paper investigated three aggregates with high alkali-reaction tendency. This paper studied concrete specimnes with including three alkali-silica reaction aggregates with accelerated inhibition test against concrete prism test. The results have been reported and appropriately discussed. However, some major revisions are proposed. The following comments could be considered in the revised manuscript:
- The English of the manuscript considerable improvements in order to fix some wordiness or grammatical problems. For instance, the sentence the abstract needs to be rearranged to be more decisive and clear. The authors did not clearly enlighten the novelty of the study; also, their incentive is not completely cover their findings in the abstract. The major points of the study and the results of the paper should be pointed out.
- The introduction needs better cohesion; it is not clear why the researchers have employed the XRD technique instead of XPS or XRF; they must provide the priority of the XRD in comparison to other typical methods. It is required to write the majority of this study (novelty points) at the end of the introduction. Also, the reason for these specific aggregates did not mention in the literature.
- The literature requires to cover more related studies in the paper. In terms of literature, there is still a need to review studies in mechanochemical issues, chemical attacks, and health monitoring. Thus, some extra-strong related references are prospered below to be used; adding these studies into the introduction would make the literature robust and comprehensive and is beneficial for the readers to find the storyline of the paper ultimately:
- "Characterizing fiber reinforced concrete incorporating zeolite and metakaolin as natural pozzolans",
- "Strength optimization of cementitious composites reinforced by carbon nanotubes and Titania nanoparticles",
- and "Utilizing Artificial Intelligence to Predict the Superplasticizer Demand of Self-Consolidating Concrete Incorporating Pumice, Slag, and Fly Ash Powders."
- Figure 7 (a) has begin with a negative value, this negative value should be explained, or if it is from the uncalibrated measurement?
- In the discussion section, the authors could include some figures of the samples after the attacks. Also, a linear regression can interpret the relationship between different chemical parameters and the bonding energy.
- The conclusion is not satisfying, and there is no evidence of the major points of the study. The conclusion should be enhanced by additional explanations about the reason for the research and the authors' intention. A potential proposal for future studies could note the capacity of the subject.
In the present form of the paper, my decision is a major revision. However, I would be glad to review the article after the authors’ revision.
Reviewer 2 Report
- The abstract should be structured and stand alone paragraph. It should start with the problem definition, challenges, work done, author's way to address them, and results or findings with the conclusion. Please rework the abstract.
- A sentence in line no. 30 is not necessary, also it makes an ambiguous statement. Please remove and start the introduction with a direct approach to the problem.
- Please avoid the abbreviation of alkali-silica reaction in the title and rather use the full form for better clarity of the manuscript's content.
- A sentence of line 42 requires reformation, "adding proper the..." is incorrect.
- Similarly, there is repetitive use of "on the other hand" is observed in the paragraphs. please rework.
- Avoid using "many countries"...please note that you are addressing the scientific communities, not countries. Hence kindly replace such words.
- An introduction must end with the work proposed and carried out. Add a connecting paragraph between the first and second sections.
- Clearly mention the research significance using a separate section.
- Please use the full form of days instead of d everywhere. Similarly use weeks or week instead of w. i.e. in figure 6.
- Section 3 represents only the comparative results with the scientific reasons for the alteration noticed. For a scientific manuscript, the causes of any change in behavior must be presented. Hence rework on the section and provide the reasons behind the changes in the standard behavior or test performance.
- Figure 14 requires more clarity on the specimen types. The photographs should consist of some scale or measurement indications always denoting the approximate size of the specimen. Rework on this aspect.
- None of the conclusions are based on the scientific clarity and interpretation of the test results. Please understand the need for a concluding remark. The conclusion should relate to the observations and scientific explanation presented in the sections of results and discussion. It is necessary to rework all sections of the manuscript completely.
- The aggregates were subjected to a specific range of temperature for the experiment. Why the impact or influence of the raised temperature on the strength properties of the aggregates has not been carried out? Or, what was the impact of the temperature rise on the different properties of aggregates? At least it must be confirmed that the proposed method does not adversely affect the aggregates or other constituents' properties.
- What is the significance of mentioning the chemical compositions of the aggregates? Where the information has been used in the explanation, result, and discussion?
- Total how many specimens were prepared? How they were get evaluated? What was the experimental program and arrangements?
- What was the mix design adopted for table 4? Which code was followed for the same? Why was the water and water-reducing was of a constant value of 1000g and 4.8?
- Why the curing setup and temperature setup are not shown anywhere?
- The values shown in the results of Figure 7 are showing the omnidirectional expansion or volumetric? Please specify in each case.
Reviewer 3 Report
The article describes rapid test method for evaluating inhibiting effectiveness of supplementary cementitious materials on ASR expansion of concrete. The article is well written. Hoverer, I have some major recommendations before further processing:
- I suggest to not using abbreviations in the Title (e.g. ASR). Please rather use full term,
- abstract is not well written. It should contain necessary information what is in this paper – not introduce to the topic,
- Introduction is too short and do not highlights the significance of the research properly. There are many citation pockets (e.g. [7-9] and many others). In my opinion each reference should be discussed and cited separately (e.g. as mentioned in [8]) and if it is not possible the unnecessary references should be deleted,
- the abbreviation alkali-silica reaction (ASR) has been defined twice in the introduction (in lines 32 and 35). In my opinion it is enough to define it once and then use the abbreviation,
- Tab. 1, tab. 2 and tab. 3 – total should give 100% (not 98.87 % like in Tab. 1). I suggest to add the column with other elements to sum up to total of 100%,
- Fig. 1 – I suggest to add the word “visual” to the caption because we have visual appearance of the aggregates,
- conclusuions should be deeper and should contain some perspectives for future research.
Reviewer 4 Report
It should be mentioned that a large enough experimental work has been done. True, in evaluating its results, I would like to see a broader discussion with the work done by other authors, and not just a statement of the facts obtained. On the other hand, there are a lot of questions that authors need to answer in order to bring the paper to the level of a submission to Materials.
Specific comments:
- Please do not use abbreviations in the Abstract and even more so in the Title without first explaining them.These abbreviations are well known to cement chemists, but they are not the only ones reading the articles.
- This section mainly cites very old literature sources. It could not have been otherwise, as it focuses on textbook truths rather than the latest scientific advances. This part of the work needs to be carefully reviewed and rewritten.
- Please do not use citation pockets (e.g. [7–9]; [11–12]; [13–15]) but rather discuss each reference individually (e.g. as described in [7]) and delete unnecessary references. All the more so as the Introduction part is very concise.
- 1. What scientific information does Figure 1 provide? All the more so as the dimensions of the stones are not marked. I suggest removing this picture from work.
- 3, line 75. “JC/T874-2009”. Since I don’t know, so I ask: here’s a standard or a technical description? Which countries? Are they published in a language understood by the general scientific community? Why it is not cited in the bibliography? Please explain and at least indicate the methods (chemical or otherwise) used to determine the chemical composition.
- 5, line 79. Please indicate the brands, manufacturers and countries of origin of all analysis instruments. Please also indicate the main setting parameters. For example: “XRD was performed on the D8 Advance diffractometer (Bruker AXS GmbH, Karlsruhe, Germany) operating at the tube voltage of 40 kV and tube current of 40 mA. The X-ray beam was filtered with Ni 0.02 mm filter to select the CuKα wavelength. Diffraction patterns were recorded in a Bragg-Brentano geometry using a fast counting detector Bruker Lynx Eye based on a silicon strip technology. The samples were scanned over the range 2θ = 3–70° at a scanning speed of 6° min-1 using a coupled two theta/theta scan type”.
- 2. The y-axis must also have units. In your case: "a.u".
- P3, line 86. “Polarized light microscopy method”. The same question as in point 6.
- What value do Figures 3, 4 and 5 add to the work? You want to see what you want. The qualitative mineral composition is determined by the XRD method. I suggest replacing these figures with a qualitative Rietveld analysis – then it would be clear what the amount of each mineral is in the raw materials.
- 5, line103. What is the specific surface area or particle size distribution of FA and BSF? Please use SI system units at work. Why only “Baogang” is written in one place and “Jiangsu Subote Co., LTD” in another.
To sum up, it is a very responsible job to come up with a new methodology and propose it to the general scientific community. Not the slightest doubt can remain in such work. Unfortunately, there is already a great deal of ambiguity and unreliable data in the introductory parts. As a rule, new methodologies are developed jointly by several institutions and their reliability is verified by independent experts. If in short, I was not convinced by this work, it does not meet the Materials level and I suggest rejecting it.
Round 2
Reviewer 1 Report
Review assignment for Materials-1667813:
I am satisfied with the changes in the revised manuscript. I recommend the paper be published.
Author Response
Thank you very much for your comments.
Thank you very much for reviewing my manuscript and opinion.
Reviewer 3 Report
No more changes are required
Author Response

(The authors gave the same response as above.)

Reviewer 4 Report
The authors took some of my comments into account. Unfortunately, the Introduction section has not been corrected enough. The information in lines 75-85 is the subject of a methodology rather than a problem analysis. This part needs to be rethought and rewritten.
Fig. 2. Since several XRD patterns are shown in this Figure, the intensity should be expressed in approximately units (a. u) or relative units (Rel. u.) and not in Counts. Please select in accordance with the Journal recommendations and rules.
Fig. 3-5. I agree that the authors know best what factual material best illustrates their claims and withdraw my request to remove these Figures from the work. Unfortunately, the authors did not take into account my remark on Rietveld's analysis and did not comment on it. I stick to my opinion and ask the authors to perform a Rietveld analysis of the samples.
I think the quality of this work still needs to be improved in my proposal – Major Revision.
